# Knockdown of *Quinolinate Phosphoribosyltransferase* Results in Decreased Salicylic Acid-Mediated Pathogen Resistance in *Arabidopsis thaliana*

**DOI:** 10.3390/ijms22168484

**Published:** 2021-08-06

**Authors:** Shengchun Li, Haiyan Ding, Yi Deng, Jiang Zhang

**Affiliations:** State Key Laboratory of Biocatalysis and Enzyme Engineering, School of Life Sciences, Hubei University, Wuhan 430062, China; inghaiyan24@163.com (H.D.); dengyi10998@163.com (Y.D.); zhangjiang@hubu.edu.cn (J.Z.)

**Keywords:** NAD, QPRT, pathogen, pathogenesis-related genes, SA, oxidative stress

## Abstract

Nicotinamide adenine dinucleotide (NAD) is a pivotal coenzyme that has emerged as a central hub linking redox equilibrium and signal transduction in living cells. The homeostasis of NAD is required for plant growth, development, and adaption to environmental stresses. Quinolinate phosphoribosyltransferase (QPRT) is a key enzyme in NAD de novo synthesis pathway. T-DNA-based disruption of *QPRT* gene is embryo lethal in *Arabidopsis thaliana*. Therefore, to investigate the function of *QPRT* in Arabidopsis, we generated transgenic plants with decreased *QPRT* using the RNA interference approach. While interference of *QPRT* gene led to an impairment of NAD biosynthesis, the *QPRT* RNAi plants did not display distinguishable phenotypes under the optimal condition in comparison with wild-type plants. Intriguingly, they exhibited enhanced sensitivity to an avirulent strain of *Pseudomonas syringae* pv. *tomato* (*Pst-avrRpt2*), which was accompanied by a reduction in salicylic acid (SA) accumulation and down-regulation of pathogenesis-related genes expression as compared with the wild type. Moreover, oxidative stress marker genes including *GSTU24*, *OXI1*, *AOX1* and *FER1* were markedly repressed in the *QPRT* RNAi plants. Taken together, these data emphasized the importance of QPRT in NAD biosynthesis and immunity defense, suggesting that decreased antibacterial immunity through the alteration of NAD status could be attributed to SA- and reactive oxygen species-dependent pathways.

## 1. Introduction

Pyridine nucleotide co-enzymes are ubiquitous in living organisms [1]. Nicotinamide adenine dinucleotide (NAD) and its phosphate derivative NADP, exist in oxidized forms (NAD^+^ or NADP^+^, respectively) or reduced forms (NADH or NADPH, respectively). They mediate a wide range of redox reactions, and thus impact virtually all metabolic reactions in the cell [2]. In addition to their functional roles in redox regulation, NADs also plays important roles in cell signaling, including intracellular Ca^2+^ signalling via NAD-derived cyclic ADP-ribose [3], translational modification of target proteins by the transfer of ADP-ribose from NAD catalyzed by poly(ADP-ribose) polymerases and mono(ADP-ribosyl)transferase [4], and epigenetic regulations by the sirtuin histone deacetylases [5,6].

In plants, NAD has also been documented to act as a signal in response to environmental stresses, including pathogen infections [7,8,9]. For instance, quinolinate-induced stimulation of intracellular NAD in Arabidopsis expressing *nadC* gene from *Escherichia coli*, which encodes quinolinate phosphoribosyltransferase (QPRT), enhances defense gene expression and resistance to diverse bacterial and fungal pathogens [10,11]. Overexpression of Arabidopsis Nudix (nucleoside diphosphates linked to some moiety X) hydrolase gene, hydrolase homolog 6 (*AtNUDT6*), encoding an ADP-ribose/NADH pyrophosphohydrolase, and disruption of *AtNUDT6* [12], *AtNUDT7* [13,14], or *AtNUDT8* [15] all lead to changes in intracellular NADH levels and salicylic acid (SA)-mediated immune signaling. Additionally, exogenous NAD(P) treatment in Arabidopsis can induce SA-dependent and -independent expression of pathogenesis-related (*PR*) genes and resistance to bacterial pathogens [16,17]. The recent identification of plasma membrane-localized NAD(P) receptors LecRK-I.8 and LecRK-VI.2 confirmed that NAD(P) act as extracellular signals [18,19].

NAD can be synthesized via both a de novo pathway and a salvage pathway in plants (Figure 1) [2,9,20]. The de novo pathway starts in plastids using aspartate as precursor. In plastids, quinolinate is produced from aspartate and dihydroxyacetone phosphate by aspartate oxidase (AO) plus quinolinate synthase (QS), and quinolinate is rapidly converted to nicotinate mononucleotide (NaMN) by QPRT. NaMN is thereafter converted to NAD in the cytosol through adenylation by NaMN adenylyltransferase (NaMNAT) and amidation by NAD synthase (NADS). *AO*, *QS*, *QPRT* and *NaMNAT* are single genes in the Arabidopsis genome. The T-DNA insertion of either of these genes exhibited developmental defects or was embryo lethal [20,21], indicating an essential role of the de novo pathway in plant growth and development. NaMNAT and NADS also participate in the NAD salvage pathway (Figure 1) [9]. Unlike the disrupted *AO* allele [20], *AO*-knockdown mutants are viable and fertile plants, but harbor impaired stomatal immunity against a coronatine-deficient strain of *Pseudomonas syringae* pv. *tomato* (*Pst*) DC3000 [22]. Arabidopsis QS consists of a quinolinate synthase (NadA) and a SufE3 domain required for incorporation of the Fe-S cluster [23]. A non-lethal mutation in the SufE domain is responsible for the early senescence phenotype of the *onset of leaf death5* (*old5*) mutant [24]. Rather than decreased NAD, this effect was associated with increased level of NAD, which was attributed to enhanced activities of the salvage pathway in the *old5* mutant [24]. Recent studies have shown that a point mutation (Q288E substitution) in the region encoding the NadA domain of the *QS* gene resulted in markedly decreased levels of NAD, and caused salt and abscisic acid (ABA) hypersensitivity in Arabidopsis [25,26]. To date, no detailed studies of *QPRT* knockdown mutants of Arabidopsis have been reported.

The NAD salvage pathway starts from nicotinamide (Figure 1), which is sequentially catalyzed by nicotinamidase (NIC), nicotinate phosphoribosyltransferase (NaPRT), NaMNAT, and NADS. Three *NIC* genes (*NIC1*, *NIC2*, and *NIC3*) have been identified in Arabidopsis [27,28]. The major NIC appears to be NIC1, and null mutant *nic1-1* shows decreased pools of pyridine nucleotides in many tissues and is hypersensitive to salt and ABA treatments, pointing to a role for the pathway in recycling and maintaining NAD pools [27]. Moreover, a 60-fold increase in nicotinamide contents in *nic1-1* Arabidopsis mutants was shown to inhibit aphid reproductive potential [29]. The *NIC2* gene is most strongly expressed in mature seeds. Increased levels of NAD was observed in *nic2-1* mutant, and this was associated with increased seed dormancy [28].

Since disruption of *QPRT* causes lethality in Arabidopsis [20], *QPRT* knockdown RNAi lines were generated to assess the effects of a reduced level of *QPRT* on pathogen infection. Decreased NAD levels in the *QPRT* RNAi plants led to enhanced pathogen sensitivity accompanied by decreased expression of *PR* genes and oxidative marker genes, and also reduced SA accumulation.

## 2. Results

### 2.1. Generation of Transgenic Plants with Decreased QPRT Expression

To assess the effects of decreased expression of *QPRT*, we generated RNAi knockdown lines by transforming wild-type Arabidopsis plants with a 343 bp fragment from cDNA in sense and antisense directions under the 35S promoter (Figure 2A). We selected two independent RNAi lines (*At*-pHY6#2 and *At*-pHY6#13) with greatly reduced *QPRT* mRNA levels (Figure 2B) and used the homologous T3 progeny of these lines in subsequent analyses (Appendix A). As expected, the *QPRT* RNAi lines showed reduced levels of NAD (Figure 2C). Moreover, a significant decrease in NADP content was observed in *At*-pHY6#13 plants (Figure 2D). Neither the NADH nor the NADPH pool was significantly affected in the RNAi lines (Figure 2C,D). However, reduced *QPRT* expression did not cause aberrant phenotypes when plants were grown under the conditions of these experiments (Appendix A).

### 2.2. Decreased QPRT Enhances Sensitivity to Pst-avrRpt2

To determine whether decreased expression of *QPRT* affects plant resistance to biotic stress, we investigated the response of the RNAi lines to an avirulent pathogen of *Pst* DC3000 harboring the elicitor avrRpt2 (*Pst-avrRpt2*). As shown Figure 3A, the growth of *Pst-avrRpt2* in the *QPRT* RNAi plants was significantly higher than that in the wild-type plants, both at 24 h post inoculation (24 hpi) and 48 hpi. To understand the mechanism by which the decreased *QPRT* expression led to enhanced disease sensitivity, we examined defense responses including transcripts of a key gene involved in SA synthesis (*isochorismate synthase 1*, *ICS1*) and *PR* genes, as well as SA accumulation. Quantitative RT-PCR analysis showed that transcripts of *ICS1*, *PR1* and *PR5* were reduced in response to *Pst-avrRpt2* in the *QPRT* RNAi plants in comparison with Col-0 at 48 hpi (Figure 3B–D). Moreover, down-regulation of these transcriptional levels coincided with reduced accumulation of total SA in the RNAi lines (Figure 3E).

### 2.3. Decreased QPRT Affects Redox Homeostasis

To test whether the observed sensitivity was accompanied by changes in NAD pools, we determined the expression level of *QPRT* gene and contents of pyridine nucleotides in the *QPRT* RNAi plants infected by *Pst-avrRpt2*. There was a relatively low level of *QPRT* expression in the RNAi lines in comparison with the wild-type plants at 48 hpi (Appendix A). Consistent with *QPRT* expression, the NAD levels were lower in the *QPRT* RNAi plants than that in Col-0 at 48 hpi, whereas much higher levels of NADH were seen in the RNAi lines (Figure 4A,B). In addition, the levels of NADP and NADPH did not differ significantly between the RNAi lines and Col-0 at 48 hpi (Figure 4C,D).

Reactive oxygen species (ROS) burst has been well established as an integral aspect of plant immunity [30,31], and NAD has been known to stimulate ROS production [11]. To further investigate overall cellular redox states, we measured changes in hydrogen peroxide (H_2_O_2_) indirectly by measuring changes in the transcript levels of four H_2_O_2_ marker genes, cytosolic glutathione *S*-transferase TAU 24 (*GSTU24*, AT1G17170) [32] and oxidative signal-inducible 1 (*OXI1*, AT3G25250) [33], mitochondrial alternative oxidase 1 (*AOX1*, AT3G22370) [34], and chloroplast *ferritin 1* (*FER1*, AT5G01600) [35]. The expression of these genes was reduced in the RNAi plants in comparison with wild-type plants at 48 hpi (Figure 5).

## 3. Discussion

Because the homozygous null alleles of *qprt* mutants of Arabidopsis are embryo-lethal [20], *QPRT* RNAi plants were generated to assess the effects of constitutively reduced level of *QPRT*. The amount of NAD was lower in the *QPRT* RNAi plants, while the NADH levels were similar to those observed in control plants under optimal conditions (Figure 2C). Pyridine nucleotide measurements in this study represent total nucleotide pools, i.e., they include free pools and those bound to proteins. It was revealed that the free NADH level can be maintained at a more constant value than oxidized NAD and total NAD [36], partially providing an explanation for the NADH levels not changing significantly in the RNAi lines. No obvious phenotype was observed in the RNAi lines under the conditions of these experiments (Appendix A).

A growing body of evidence suggests that NAD plays a crucial role in plant immunity [10,11,16,22,37]. Notably, overexpression of the bacterial NAD biosynthesis gene *nadC*, which increased intracellular NAD levels with the addition of quinolinate, resulted in heightened resistance to avirulent pathogen *Pst-AvrRpm1,* but not to virulent strain *Pst* DC3000 [10]. In the current study, we tested whether a deficient NAD synthesis pathway in *QPRT* RNAi lines would have increased susceptibility to an avirulent bacterial strain. As expected, after infection with the avirulent avrRpt2-containing *Pst* DC3000 strain (*Pst-avrRpt2*), the *QPRT* RNAi plants exhibited increased bacterial proliferation compared with Col-0. The enhanced sensitivity to *Pst-avrRpt2* was associated with suppressed expression of the defense-related genes (*ICS1*, *PR1* and *PR5*), and this was consistent with decreased SA accumulation (Figure 3). In addition, the NAD levels were decreased following the infection with *Pst-avrRpt2*, whereas the NADH levels were increased (Figure 2C; Figure 4A,B). A much higher level of NADH was observed in the RNAi lines in comparison with Col-0 after *Pst-avrRpt2* infection. The observed alterations of NAD(H) pools led to a lower NAD/NADH ratio in the RNAi lines than that in wild-type plants. Indeed, the physiologically relevant ratio of NAD to NADH is generally high, favoring hydride transfer from a substrate to NAD to form NADH in plant cells [38]. The results of the current study support that QPRT enzyme is critical for the steady state of NAD and the homeostasis of NAD/NADH in Arabidopsis.

The cytosolic *GSTU24* gene is known to be H_2_O_2_ inducible [32,39] and up-regulated by *Pst-avrRpt2* [40,41]. OXI1, a serine/threonine kinase, activates the MAPK3/6 cascade in an H_2_O_2_-dependent manner, which has been implicated in some pathogen responses and root hair development [33,42]. *AOX1* encodes a mitochondrial alternative oxidase that has been shown to be induced by *Pst* attack [43] and links mitochondrial ROS to cell death [44,45]. *FER1,* encoding chloroplast-localized ferritins, has been shown to play an important role in keeping Fe^2+^ levels at a minimum upon increased intracellular H_2_O_2_ [35,46]. In the current study, reduced expression of four H_2_O_2_ marker genes suggests redox perturbations occurred in different subcellular compartments in *QPRT* RNAi plants. Given that the cytosol is the site of the final step of NAD biosynthesis (Figure 1), specific transport proteins are required to shuttle NAD across intracellular membranes for organellar import of NAD into chloroplasts, mitochondria and peroxisomes [47,48,49,50,51]. NAD transport could influence cellular redox balance, thus may lead to redox perturbations in different subcellular compartments of *QPRT* RNAi plants.

Collectively, these results indicate that QPRT is important in NAD biosynthesis, and contributes to plant resistance through SA- and ROS-dependent pathways. Additionally, the availability of the *QPRT* RNAi lines allow future detailed studies on the role of NAD metabolism in many aspects of plant biology including, but not limited to, immunity, development, signaling, and biosynthesis [8,9,38].

## 4. Materials and Methods

### 4.1. Plant Materials and Growth Condition

The RNAi lines used in this study were derived from the wild-type *Arabidopsis thaliana* Columbia (Col-0) ecotype. Plants were cultivated in controlled growth chambers in a photoperiod of 8 h light/16 h dark, an irradiance of 100 μmol·m^−2^s^−1^ at leaf level, at temperatures of 22 °C day/20 °C night and 75–80% humidity.

### 4.2. Generation of RNAi Plants

To obtain RNAi plants, partial coding region for *QPRT* (AT2G01350) was amplified with RiQPRT-Fwd1/RiQPRT-Rev1 and RiQPRT-Fwd2/RiQPRT-Rev2 primers (Appendix A). The primers harbor a short homologous sequence (underlined) to pHELLSGATE8 vector. Two *QPRT* PCR products were then cloned into the pHELLSGATE8 vector via homologous recombination between the XhoI-XhoI sites in the sense orientation and the XbaI-XbaI sites in the antisense orientation to generate RNAi construct pHY6 (Figure 2A) [52]. The construct was transformed into the *Agrobacterium tumefaciens* GV3101 strain, which was then used to transform Arabidopsis Col-0 using the floral dip method [53]. Transgenic plants were selected on Murashige and Skoog (MS) media containing 50mg·L^−1^ kanamycin. Single insertion lines were selected by 3:1 segregation on selection media.

### 4.3. Pathogen Test

The avirulence strain of *Pseudomonas syringae* pv. *tomato* DC3000 carrying the avirulent gene *avrRpt2* (*Pst-avrRpt2*; a gift from Prof. Jianmin Zhou, Institute of Genetics and Developmental Biology, Chinese Academy of Sciences) was used for resistance tests with a medium titer of 5 × 10^5^ colony-forming units (cfu) mL^−1^. Whole leaves of 5-week-old plants were infiltrated with a 1-mL needleless syringe in the middle of the photoperiod. Six to eight leaves were made by pooling two leaf discs (φ = 0.45 cm each) harvested from different inoculated leaves of different plants. Bacterial growth was examined by grinding leaf discs in 400 mL of water, plating appropriate dilutions on solid King’s B medium with 100 mg·L^−1^ rifampicin and 25 mg·L^−1^ kanamycin, and quantifying colony numbers. Bacterial cfu were counted at 0, 24 and 48 hpi. Experiments were repeated three times with similar results.

### 4.4. RNA Isolation and Quantitative Reverse Transcription-PCR

Total RNA was isolated from leaf samples using RNAiso Plus Reagent (Takara, Dalian, China) according to the manufacturer’s instructions. Hifair^®^ Ⅱ 1st Strand cDNA Synthesis Kit with gDNA digester (Yeasen, China) was used to synthesize cDNA. Quantitative reverse transcription-PCR (qRT-PCR) was performed on Bio-Rad CFX Connect Real-Time System (Bio-Rad, USA) using the SYBR^®^ Premix Ex Taq™ II (Tli RNaseH Plus) kit (Takara, Dalian, China). The *ACTIN2* gene was used as an internal control to normalize for specific gene expression in the samples. Each treatment was replicated with at least three independent biological sample preparations. Quantitative analysis of gene expression was performed using the 2^−ΔΔCT^ method [54]. Gene primers are listed in Appendix A.

### 4.5. Metabolite Measurements

Reduced and oxidized forms of NAD and NADP were determined by spectrophotometric assays, as previously described [55,56]. Total SA concentration was measured by high-performance liquid chromatography-fluorescence according to Langlois-Meurinne et al. (2005) [57].

### 4.6. Statistical Analysis

Statistical significance was determined using the Student’s *t*-test. A *p* value of <0.05 was considered significant and indicated by an asterisk in the Figures.

## Figures and Tables

**Figure 1 ijms-22-08484-f001:**
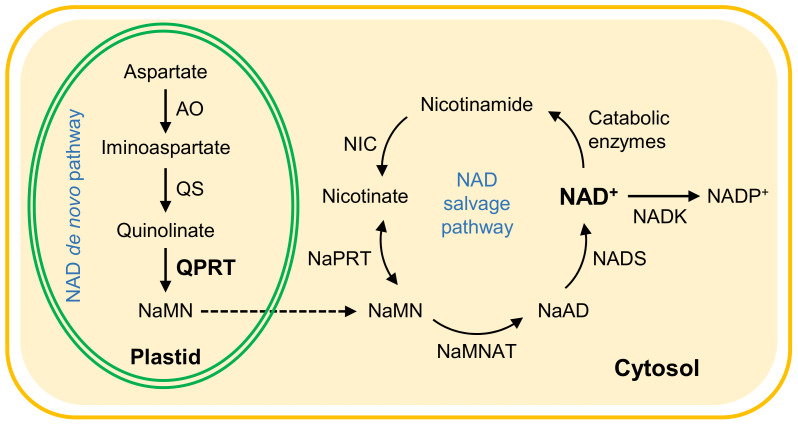
Schematic representation of NAD biosynthesis pathway in plants. The de novo biosynthesis of NAD starts from aspartate in the plastid. Abbreviations: AO, aspartate oxidase; NaAD, nicotinate adenine dinucleotide; NADK, NAD kinase; NADS, NAD synthetase; NaMN(AT), nicotinate mononucleotide (adenylyltransferase); NaPRT, nicotinate phosphoribosyltransferase; NIC, nicotinamidase; QPRT, quinolinate phosphoribosyltransferase; QS, quinolinate synthase.

**Figure 2 ijms-22-08484-f002:**
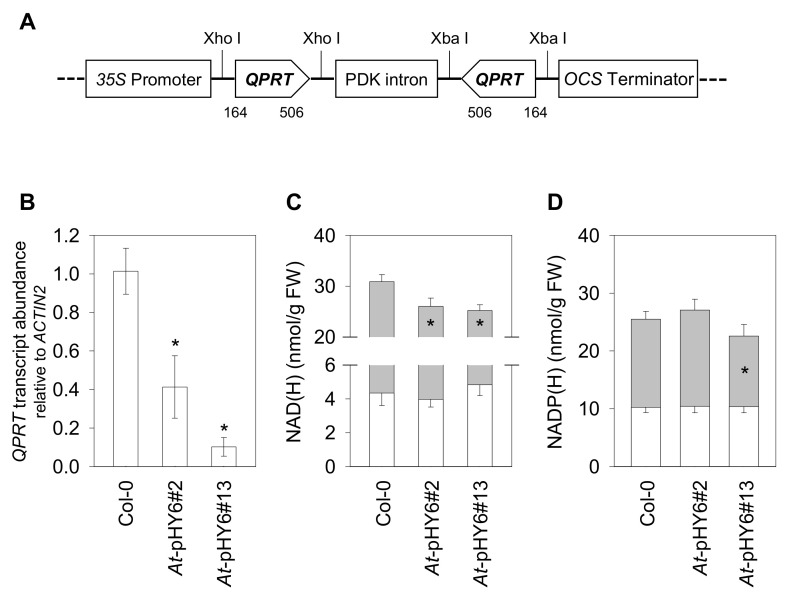
Generation of *QPRT* RNAi lines and determination of pyridine nucleotides contents in Arabidopsis. (**A**) The RNAi construct pHY6 contained a 35S promoter, a sense fragment of *QPRT* cDNA from 164 to 506, the PDK intron, the *QPRT* fragment in antisense orientation, and an *OCS* terminator. (**B**) *QPRT* mRNA levels analyzed by qRT-PCR. Contents of NAD(H) (**C**) and NADP(H) (**D**) were quantified. White bars, reduced forms; black bars, oxidized forms. Data are means ± SE of three biological replicates. Asterisk indicates difference relative to Col-0 for each form at * *p* < 0.05.

**Figure 3 ijms-22-08484-f003:**
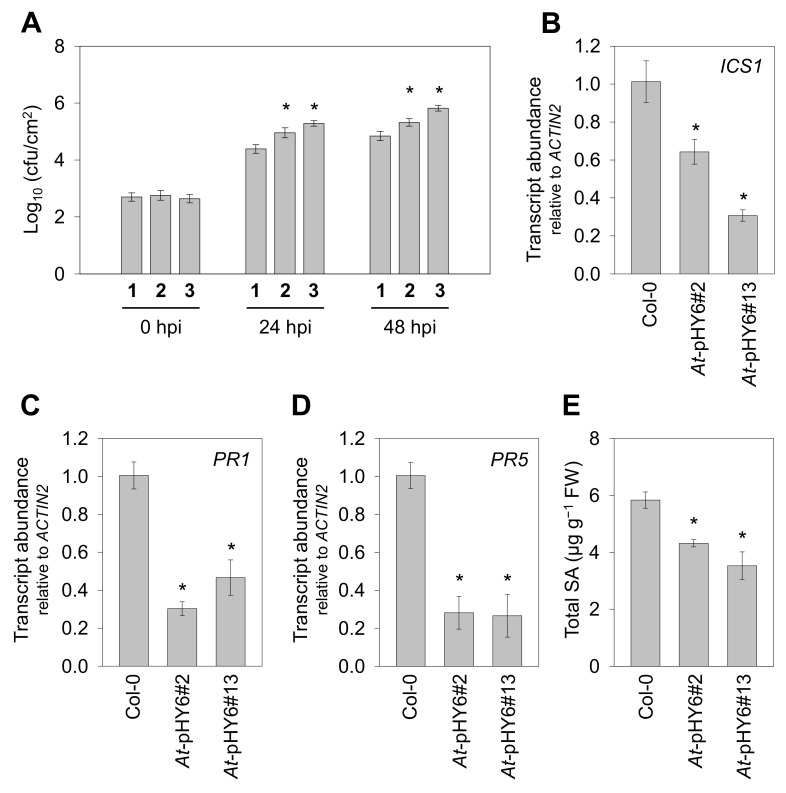
Reduced transcript levels of *QPRT* are associated with increased susceptibility. (**A**) The *QPRT* RNAi plants showed enhanced sensitivity to *Pst-avrRpt2* compared with Col-0. Leaves from two 5-week-old independent lines of *QPRT* RNAi or Col-0 were inoculated with *Pst-avrRpt2* at a concentration of 5 × 10^5^ cfu·mL^−1^. Bacterial growth was measured at 0, 24, and 48 hpi. **1**, Col-0; **2**, *At*-pHY6#2; **3**, *At*-pHY6#13. Expression of *ICS1* (**B**), *PR1* (**C**) and *PR5* (**D**) was measured by qRT-PCR in Arabidopsis challenged with *Pst-avrRpt2* at 48 hpi. (**E**) Quantification of SA in pathogen-challenged plants at 48 hpi. Means ± SE of three repetitions are shown. Asterisk indicates means are significantly different from those of Col-0 at *p* < 0.05.

**Figure 4 ijms-22-08484-f004:**
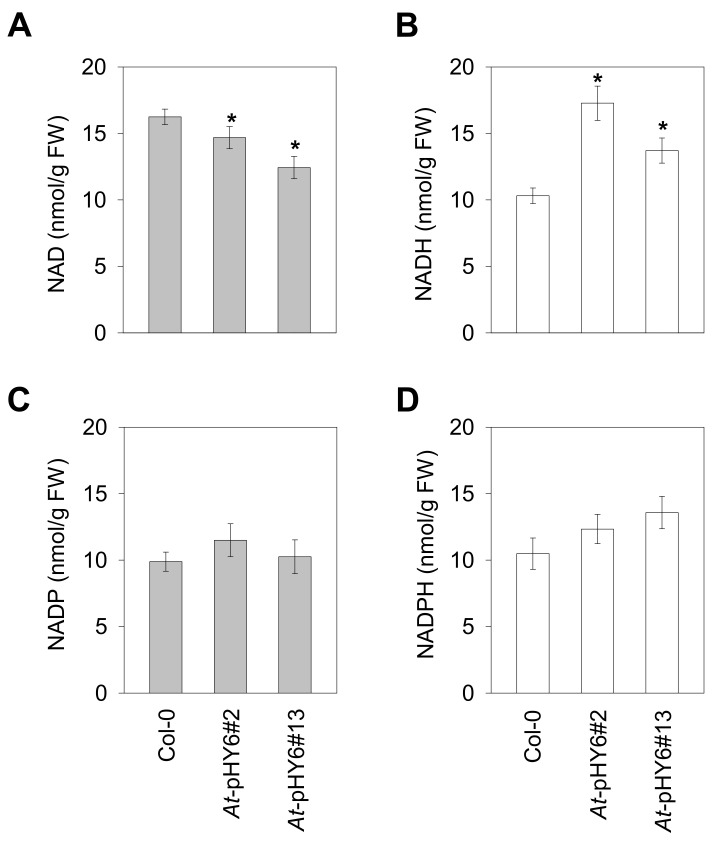
Contents of pyridine nucleotides in Col-0 and RNAi plants upon *Pst-avrRpt2* infection. Levels of NAD (**A**), NADH (**B**), NADP (**C**) and NADPH (**D**) in bacterium-challenged plants at 48 hpi were determined. Data are means ± SE of four independent extracts. Asterisk indicates significant differences from Col-0. * *p* < 0.05.

**Figure 5 ijms-22-08484-f005:**
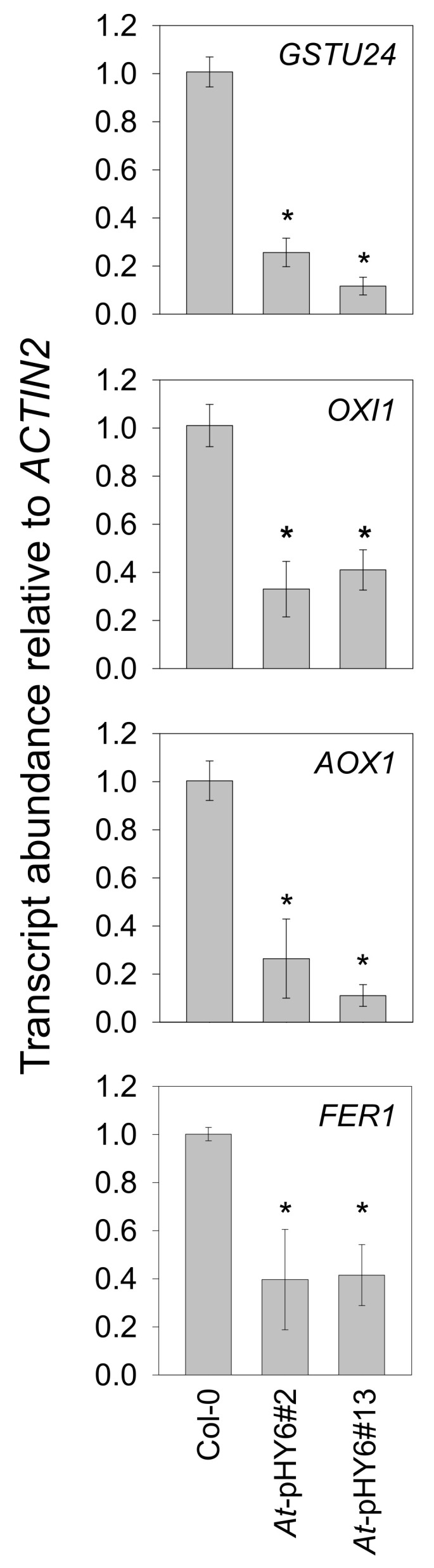
Transcript abundance of oxidative stress marker genes were suppressed in the RNAi plants compared with Col-0 at 48 hpi. Transcriptional expression is shown relative to the reference gene *ACTIN2*. Values shown indicate means of four biological replicates. Asterisks indicate that values are significantly different from those of Col-0. * *p* < 0.05.

## Data Availability

This study did not report any data.

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
