# Peer review of "Knockdown of Quinolinate Phosphoribosyltransferase Results in Decreased Salicylic Acid-Mediated Pathogen Resistance in Arabidopsis thaliana"

_ijms, 2021, doi:10.3390/ijms22168484_

Round 1

Reviewer 1 Report

Although pretty good, grammar needs improvement and, in several places, the scientific language (especially phytopathological, microbiological and statistical) needs to be corrected. In a few places, more extensive rewriting is required. There’s not much to the Discussion and it may need to be rewritten.

“Knockout” is considered slang or jargon and the proper word is “disruption.” I don’t know about “knockdown”; it seems using “RNAi” or “reduced transcript levels” would be most appropriate. It’s up to the editor, how they want to present.

A lot of their results are associative. (A lot of science is.) Authors have not investigated exactly what causes their results. Therefore, they need to be cautious as to how they describe their results.

Authors used “repressed” to describe reduced levels of transcripts. This is imprecise as authors did not investigate the mechanism causing the reduced transcript levels (repression implies that a factor or factors are preventing induction of the gene).

Authors are using an avirulent strain of Pseudomonas syringae pv tomato. Throughout, authors need to be careful to state that it is avirulent on Col-0.

Figure legends: “using a plate-reader assay” is very general and does not say much. Suggest briefly describing how the assay was done or provide a reference.

Specific comments/editorial suggestions:

Abstract:

Lines 17 – 18: replace “phenotype under optimal condition” with “phenotypes under the optimal condition”

Lines 19 – 20: insert “as compared with the wild-type” after “expression”

Introduction:

Lines 29 – 30: change “as oxidized form…or reduced form…” to “in oxidized…or reduced forms…”

Line 31: remove “on”

Line 33: the “2” in “Ca2+” should be superscript

Line 44: change “knocking-out” with “disruption of”

Line 48: change “pathogen” to “pathogens”

Line 58: change line to “Arabidopsis genome. The T-DNA insertion mutagenesis of any one of these genes had resulted in developmental…”

Line 61: change “knockout” to “disrupted”

Line 62: change “harbors” to “harbor”

Line 68: insert “the” before “old5”

Line 70: change “has” to “resulted in”

Line 75: insert “the” before “plastid”

Line 83: replace “recycling pathway in” to “pathway in recycling and”

Line 84: “nic-1-1” needs correcting

Line 87: replace “increase in” with “increased”

Line 88: replace “knockout” with “disruption”

Line 89: insert “a” between “of” and “reduced”

Results:

Line 100: change “level” to “levels”

Line 104: replace “phenotype” with “phenotypes”; replace “in our growth chamber” to “under conditions of these experiments”

Line 109: remove “using a plate-reader assay”

Line 113: replace “plants’” with “plant”

Line 114: insert “of” before “Pst-”

Line 116: replace “at both” with “both at”

Lines 119 – 120: replace “and” with “as well as”; remove “as well”

Line 123: replace “lower” with “reduced”

Line 126: replace “Knockdown of QRTP caused enhanced sensitivity” with “Reduced transcript levels of QRTP is associated with increased susceptibility”

Line 131: replace “Data are means +/- SE of three repeats” with “Means +/- SE of three repetitions are shown” -how many plants per repetition?; replace “significant differences from Col-0 *P < 0.05” with “means are significantly different from those of Col-0 at P < 0.05”

Lines 133 – 137: This paragraph is confusing and needs to be rewritten.

Lines 138 – 139: Change to “To further investigate overall cellular redox states, we measured changes in the…”

Line 140: remove “including”

Line 143: change “repressed” to “reduced”

Line 146: figure title needs to be rewritten and improved

Line 147: change “pathogen” to “bacterium”; remove “using a plate-reader assay”

Line 152: change “Asterisk indicates significant difference from” to “Asterisks indicate that values are significantly different from those of”

Discussion: The discussion seems brief and the writing, language and grammar needs quite a bit of improvement

Lines 156 – 163: This paragraph needs to be entirely rewritten. It does not highlight the current research.

Lines 164 – 165: Authors could start the discussion here; change “Since” to “Because”; change “bacterial pathogen” to “the avirulent bacterial strain P. syringae pv tomato”

Lines 166 – 168: Change “knockdown” to “reduced expression”; change “level” to “levels”; change “in our growth condition” to “under conditions of these experiments”

Line 170: change “reflected by” to “associated with”

Lines 173 – 175: Sentence needs to be rewritten for grammar and the fact the strain is avirulent on Col-0.

Lines 185 – 186: replace “Suppressed” with “In the current study, reduced”

Lines 188 – 189: change “transporter” to “transporters”; replace “probably explains the” with “may result in”; replace “from” with “in”; replace “in the” with “of”

Lines 190 – 192: change “our results emphasized” with “these results indicate”; replace “made an” with “is”; replace “contribution to” with “in”; replace “confers” with “contributes”; replace “defenses” with “pathways.”

Materials and Methods:

Line 195: Change “All of” to “The”

Line 196: remove “a”

Line 197: insert “at” before “temperatures”

Lines 206 – 207: change to: “The construct was transformed into the Agrobacterium tumefaciens GV3101 strain which was then used to transform Arabidopsis Col-0 using the floral dip method.”

Line 211: change “avirulent” to “avirulence”

Line 218: How many plants per repetition?

Line 228: change “were” to “are”

Line 238: replace “marked” with “are indicated”

Reviewer 2 Report

The publication titled “Knockdown of quinolinate phosphoribosyltransferase results in decreased salicylic acid –mediated pathogen resistance in Arabidopsis thaliana” is about knockdown the QPRT gene in Arabidopsis and verifying the response of mutant plants to pathogen infection. Researchers in their study used two transgenic lines in which different levels of expression of the studied gene were determined. Experiments were also carried out to determine the expression of genes related to the response to the tested biotic stress. The studies have been logically planned, and the required repetitions and controls are maintained. The necessary statistical analyzes were also performed. The work is correct, as well as discussion.

I have a few minor comments:

- what was the expression level of the QPRT gene after pathogen inoculation?

- figure 2 - please explain the lack of differences in the reduced forms (white bars fig 2)

- figure 3 - the different colors of bars are confusing.

Only chart A has 3 colors, every other chart has bars of the same color (chart E also has bars in different shade)

- why analogous experiments were not performed for the typical Arabidopsis pathogen instead of the avirulent pathogen

- Figure 4 and 5 - the description is correct but the graphs are changed

- Line 71 - QPRT instead of QRPT

Round 2

Reviewer 1 Report

The manuscript by Li et al (formerly Ding et al) has improved quite a bit. Some parts, particularly the Discussion, need some improvement with English sentence structure and clarification of authors' meaning. Also, the impact could be improved if authors use more correct statistical and experimental design language.

Specific comments:

Results:

Line 143: remove "to"

Line 165: replace "Relatively low-level QRPT expression was maintained" with "There was a relatively low level of QRPT expression"

Line 167: replace "In consistent" with "Consistent"

Line 172: replace "well been" with "been well"

Line 173: replace "was report" with "has been known"

Discussion

Lines 211 - 212: replace "this may partially explain that" with "providing a partial explanation for the"; replace "did not change greatly" with "not changing significantly" (but only if true); replace "Intriguingly, no" with "No"

Line 217: replace "heightens" with "resulted in heightened"

Line 218: replace "Therefore, we are interested in testing" with "In the current study, we tested"

Line 219: replace "correlate with an" with "have"

Line 220: replace "Expectedly" with "As expected"

Line 223: replace "a series of" with "the"

Line 224 - 225: replace "the decrease in" with "decreased"

Line 227: replace "Particularly," with "A"

Line 228 - 229: I suggest you move this sentence to after the next one. Also, please make it clearer that this is what is normally observed in wild-type plants.

Line 231: replace "These results" with "The results of the current study"

Line 233: insert "known to be" between "is" and "H2O2"; remove "is" from between "and" and "up-"

Line 235: replace "is" with "has been"

Line 236: replace "is" with "has been shown to be"

Line 238: replace "plays" with "has been shown to play"

Line 239: remove "concentration of"

Line 244 - 245: remove period; replace "in plants," with "for"; replace "for the chloroplast, the" with "into chloroplasts"; replace "the peroxisome has been characterized" with "peroxisomes"

Line 247: insert "subcellular" before "compartments" or something else to make it clearer what authors mean.

Lines 252 - 254: "immunity" is not a cellular response but cellular responses contribute to immunity. This sentence needs to be rewritten and made much more precise.

Materials and Methods:

Line 261: replace "in" with "with"

Line 314: remove "are"

Figure and Supplemental legends:

Line 589 & 614: "Repeats" is vague. Do authors mean replicates (for individual plants) or repetitions (for entire assays)?
